# Silver Nanoparticles–Polyethyleneimine-Based Coatings with Antiviral Activity against SARS-CoV-2: A New Method to Functionalize Filtration Media

**DOI:** 10.3390/ma15144742

**Published:** 2022-07-06

**Authors:** Marta Baselga, Iratxe Uranga-Murillo, Diego de Miguel, Maykel Arias, Victor Sebastián, Julián Pardo, Manuel Arruebo

**Affiliations:** 1Institute for Health Research Aragon (IIS Aragón), 50009 Zaragoza, Spain; iratxe.um@gmail.com (I.U.-M.); diego_demiguel@hotmail.com (D.d.M.); maykelariascabrero@gmail.com (M.A.); victorse@unizar.es (V.S.); pardojim@unizar.es (J.P.); arruebom@unizar.es (M.A.); 2Department of Microbiology, Pediatrics, Radiology and Public Health, Facultad de Medicina, University of Zaragoza, 50009 Zaragoza, Spain; 3Networking Research Center on Infectious Diseases, CIBERINFEC, 28029 Madrid, Spain; 4Instituto de Nanociencia y Materiales de Aragón (INMA), CSIC-Universidad de Zaragoza, 50009 Zaragoza, Spain; 5Department of Chemical Engineering, Campus Río Ebro–Edificio I+D, University of Zaragoza, 50018 Zaragoza, Spain; 6Networking Research Center on Bioengineering, Biomaterials and Nanomedicine, CIBER-BBN, 28029 Madrid, Spain

**Keywords:** silver nanoparticles, polyethyleneimine, SARS-CoV-2, antiviral coating, facemask, filter

## Abstract

The use of face masks and air purification systems has been key to curbing the transmission of SARS-CoV-2 aerosols in the context of the current COVID-19 pandemic. However, some masks or air conditioning filtration systems are designed to remove large airborne particles or bacteria from the air, being limited their effectiveness against SARS-CoV-2. Continuous research has been aimed at improving the performance of filter materials through nanotechnology. This article presents a new low-cost method based on electrostatic forces and coordination complex formation to generate antiviral coatings on filter materials using silver nanoparticles and polyethyleneimine. Initially, the AgNPs synthesis procedure was optimized until reaching a particle size of 6.2 ± 2.6 nm, promoting a fast ionic silver release due to its reduced size, obtaining a stable colloid over time and having reduced size polydispersity. The stability of the binding of the AgNPs to the fibers was corroborated using polypropylene, polyester-viscose, and polypropylene-glass spunbond mats as substrates, obtaining very low amounts of detached AgNPs in all cases. Under simulated operational conditions, a material loss less than 1% of nanostructured silver was measured. SEM micrographs demonstrated high silver distribution homogeneity on the polymer fibers. The antiviral coatings were tested against SARS-CoV-2, obtaining inactivation yields greater than 99.9%. We believe our results will be beneficial in the fight against the current COVID-19 pandemic and in controlling other infectious airborne pathogens.

## 1. Introduction

The pandemic caused by the severe acute respiratory syndrome coronavirus 2 (SARS-CoV-2) has been characterized by a rapid spread between populations due to its high efficiency of viral transmission [1]. The global acceptance of the spread of COVID-19 by aerosols improved a preventive approach, including the mandatory use of filter half masks (KN95) or surgical masks in most countries and the purification of indoor air using filtration systems. It has been reported that the use of masks has reduced by up to 70% the chances of potential infections caused by SARS-CoV-2 [2,3,4]. However, the efficiency of surgical masks is compromised against submicron particle sizes [5], including SARS-CoV-2-loaded bioaerosols [6]. Particle leaks between 60% and 80% have been reported with surgical masks as personal protective equipment, which could be explained mainly by the material filtering ability and the poor facial fitting [7,8,9,10]. Half-masks are theoretically designed to provide a perfect facial fit, although in practice this is questionable [11,12,13]. A Centers for Disease Control and Prevention (CDC) report evaluating 21 KN95 masks on 25 volunteers revealed leaks between 6% and 88% of aerosolized particles [14]. In agreement with these results, Park et al. [15] reported leaks up to 21.1% in 3M KN95 masks and up to 73.3% in KF94 masks of Korean origin.

Research efforts towards developing effective filter systems, both as personal protective equipment (PPE) and in air conditioning systems, in the fight against COVID-19 have been directed towards the incorporation of materials with antiviral and antimicrobial properties. Mainly derived from nanotechnology, nanostructured metallic compounds have been the object of special attention in the virucidal functionalization of masks and filters. Examples of those materials include silver, graphene, copper, and zinc [16]. Specifically, silver has been incorporated into various surfaces due to its historical use as an antimicrobial agent [17] and its approval by the FDA in several drugs and devices used in clinical settings, including their use in different antimicrobial medical devices (e.g., catheters), wound dressings, and medical textiles [18]. Specifically, nanoparticulated silver (AgNP) shows an obvious advantage over its analog on a metric scale due to its high surface-to-volume ratio and easy fabrication [19,20]. AgNPs act as a reservoir of ionic silver; the nanoparticles oxidize, releasing the antimicrobial ion while leaving the nanoparticles surface having available chemisorbed ionic silver and being the remaining core prone to successive oxidation cycles. The potential antimicrobial effect depends, among other factors, on the physicochemical parameters that NPs present, including size, shape, surface charge, surface functionalization, specific surface area, concentration, and colloidal oxidation state [21]. Although its ability to destroy microorganisms has been widely demonstrated against gram-negative and gram-positive bacteria [22,23,24,25], its virus inactivation ability has been less reported [26,27].

Regarding the COVID-19 pandemic, the interaction of nanosilver with the responsible encapsulated virus has been studied in-depth, proposing two different theories. On the one hand, (1) AgNPs bind to the virus capsid, inhibiting its binding ability to receptor cells. On the other hand, (2) AgNPs bind to viral RNA, inhibiting its replication in host cells. The combined hypothesis of the binding of AgNPs to the spike protein of the SARS-CoV-2 virus at the lipidic envelope and the generation of a hostile environment in the respiratory epithelium, associated with a pH decrease due to the release of silver ions, has also been reported [28]. Although the antiviral mechanism is unknown with precision, colloidal silver has shown high efficiency in the inactivation of the SARS-CoV-2 virus [29,30], presenting better antiviral behavior than other nanostructured metals. Specifically, Merkl et al. [31] reported that silver nanoparticles reduced 98% of the SARS-CoV-2 viral load after 2 h, while copper nanoparticles achieved only 76% reduction and zinc nanoparticles did not show antiviral efficacy. The antimicrobial efficiency of AgNPs has been also reported once incorporated into different filter materials against bacteria and viruses [17,32,33]. Even though several models of masks that incorporate nanosilver are currently marketed, a recent report points out that only ~50% of commercial masks that advertise the incorporation of silver nanoparticles had substantiated antiviral and antibacterial claims [18]. An increase in the antimicrobial effectiveness of nanostructured silver has been demonstrated in numerous research articles when combined with other antimicrobial agents, such as carbon nanotubes [34,35,36], carbon nitride [37], chitosan [38], or polyethyleneimine (PEI) [39]. Specifically, this last conjugation has shown excellent antimicrobial activity against *S. aureus* and *K. pneumoniae* [39].

This paper reports a new, easy, and affordable method to fabricate a coating based on silver nanoparticles. AgNPs could be toxic to the humans at high doses due to their oxidative stress generation, DNA damage, and cytokine induction production. Consequently, a stable chemical bonding is essential to ensure the stability of the particles on the fibers and to avoid any potential detachment towards the respiratory tract when incorporated in face masks or in other filtration systems. Polyethyleneimine is a polycationic polymer made from ethyleneimine monomers. By suspending negatively charged polymeric fibers in this polymer, their charge is reversed, becoming positively charged. A strong supramolecular interaction by electrostatic interaction with negatively charged AgNPs is then guaranteed. In addition, the functional amine groups of PEI act as ligands and donate an unshared electron pair to Ag to create a coordinate covalent bond which further improves their strong attachment [40]. As a preventive strategy against the spread of SARS-CoV-2, the method reported herein has five key aspects: (1) industrial viability because of its simplicity, (2) high antiviral efficiency, (3) stable union on different polymeric filtering media, (4) homogeneity in the coating, and (5) preservation of the filtering and breathability properties of the original material. In this work, the coating was applied on different polymeric substrates based on fibers of variable diameters composed of polypropylene, polyester-viscose, and polypropylene-glass, corresponding to surgical masks, hygienic masks, and air conditioning filters of medium and high efficiency of filtration, respectively. The stability of the binding of AgNPs on the polymeric fibers was explored using different techniques. Finally, the inactivation efficiency against the SARS-CoV-2 virus was validated by using a viral isolate obtained from a COVID-19 patient.

## 2. Materials and Methods

### 2.1. Synthesis Materials and Procedures

The synthesis of silver nanoparticles was carried out by slightly modifying the method of Goli et al. [41] (Table 1) to obtain the smallest possible particle diameter to facilitate oxidation. Briefly, an aqueous solution of 1.0 mM silver nitrate (AgNO_3_) and 1.0 mM tribasic sodium citrate dihydrate (TSC) were dissolved and kept under vigorous stirring (~1250 rpm) for 5 min. Subsequently, 1.2 mM sodium borohydride (NaBH_4_) was added dropwise, and the mixture was stirred for 1 h. Finally, the colloidal dispersion of negatively charged AgNPs was washed by centrifugation at 13,000 rpm for 15 min and at 15 °C. The supernatant was removed, and the pellet redispersed in Milli-Q water until the desired colloidal concentration was achieved (Figure 1). AgNO_3_ (CAS 7761-88-8), TSC (CAS 6132-04-3), and NaBH_4_ (CAS 16940-66-2) were purchased from Sigma-Aldrich (Darmstadt, Germany) and used as received.

During the optimization of the synthesis method, process parameters such as stirring speed, solute concentration, and NaBH_4_ reagent mixing mode were varied, obtaining six differentiated syntheses (Table 1). It was decided to select the smallest particle size possible to increase the NPs’ virucidal activity, considering that the smaller the size, the faster the oxidation rate and the consequent ionic silver release [22,42].

### 2.2. Nanoparticle Attachment to Polymeric Fibers

The coated polymeric substrates were based on polypropylene, polyester-viscose, and polypropylene combined with fiberglass. Specifically, we used spunbond fabric (polypropylene) from a surgical mask, spunlace fabric (polypropylene) from a surgical mask, filter material (polypropylene) from an F9 air conditioning filter, viscose polyester-based fabric from a hygienic mask, and combined polypropylene-glass fibers from a HEPA H13 filter. The hygienic mask fabric was obtained from Ditex SL (Zaragoza, Spain). The H13 and F9 filters were kindly provided by Camfil España SA (Madrid, Spain), and the surgical masks were purchased at a local pharmacy.

Polyethyleneimine (CAS 904759) purchased from Sigma-Aldrich (Darmstadt, Germany) was used for the chemical bonding process. Briefly, PEI was dissolved in an aqueous solution at variable concentrations either 5 or 10 mg/mL. The fabric samples were immersed in the solution for 30 min, dried in the open air for 10 min, and subsequently washed with deionized water to remove all PEI in excess. After washing, the samples were suspended in a colloidal dispersion of AgNPs at 1, 2.5, and 5 mg/mL in an amount not less than 0.7 mL/cm^2^ for 1 h at room temperature. Finally, the samples were retrieved, dried at room temperature for 10 min, and thoroughly washed with deionized water to detach potential non-adhered nanoparticles.

### 2.3. Particle Size Measurements

Morphologic characterization was carried out using a T20-FEI transmission electron microscope (FEI Company, Hillsboro, OR, USA). TEM samples were prepared by depositing 50 µL of Milli-Q water dispersed colloid on a formvar-coated copper TEM grid and dried for at least 2 h. The nanoparticle size distribution histograms were calculated by measuring the particle diameters from the TEM images with the ImageJ software (National Institutes of Health, Bethesda, MD, USA, v1.53) and using basic statistical analysis (*n* = 100).

### 2.4. Ultraviolet-Visible Spectroscopy (UV-Vis)

The extinction spectra of the resulting silver colloids were recorded using a Jasco V670 UV-Vis spectrophotometer (JASCO International Co. Ltd., Tokyo, Japan) at the maximum absorbance around 400 nm. Aliquots of the analyzed samples were obtained at different times (1, 3, 5, and 7 days). Monodispersion in the particle sizes was evaluated by direct observation of the amplitude and the full width at half the maximum of the absorbance peak. Over a week, the stability over time was analyzed based on the interposition of the curves of the absorbance peaks obtained.

### 2.5. Scanning Electron Microscopy with Energy-Dispersive X-ray Analysis (SEM/EDX)

The morphology of the AgNPs-PEI coated fibers was observed using a CSEM-FEG Inspect 50 Field Emission Scanning Electron Microscope (SEM) (FEI Company, Hillsboro, OR, USA) at a high vacuum with an acceleration voltage of 10 kV. For sample preparation, a sample of the corresponding resulting coating was placed on a slide, fixed with carbon tape to a SEM microscope holder, and air-dried.

### 2.6. Bonding Stability Evaluation Using Microwave Plasma Atomic Emission Spectroscopy

Quantifiable potential particle detachment analyses were performed using MP-AES (4100 MP-AES, Agilent Technologies, Santa Clara, USA). These tests were carried out on the coated polymeric substrates (polypropylene fibers, spunbond, and surgical masks) to test the stability of the coating against different boundary conditions. For the application of mechanical stress, the coated samples were cut out and initially weighed and weighed again after the test to evaluate the mass loss by mass balance. The samples were tested using two complementary techniques: (a) the quantification of the initial amount of AgNPs impregnated in the fabric and (b) the quantification of the AgNPs released after sonication. Those analyses were carried out from digestions with aqua regia (HCl, 37%: HNO_3_, 67%, 1:5). In analysis (1), fabric samples having Ag-NPs impregnated were immersed in 1 mL of aqua regia for 30 min. Once digested, they were diluted in 5 mL of Milli-Q water and filtered through 0.2-micron filters. On the other hand, in analysis (2) the supernatant collected after 30 min of sonication (0.5 mL) was digested in 1 mL of aqua regia. Subsequently, it was diluted in 5 mL of Milli-Q and filtered through a 0.2-micron filter. Boundary conditions and the sonication times were varied. Reference sample corresponds to the aqua regia digestion of the fabric samples coated with AgNPs after 20 s of sonication in Milli-Q water used as base value. In order to standardize the results obtained, the samples were measured and weighed after carrying out the tests. The boundary conditions are described in Table 1. Atomic emission measurements were carried out by taking three characteristic peaks of silver (at 328.068, 338.289, and 546.549 nm). The calibration regressions obtained with the standards made were 0.99895, 0.99898, and 0.99907, respectively.

Additionally, the determination of the potential AgNPs release from the modified filter media was evaluated by mass balance using a precision microbalance (WLC X2 Radwag, Radom, Poland) before and after being subjected to 30 successive manual washings at room temperature in Milli-Q water. The cleaning period was, at least, during 10 s each washing using a spunbond surgical mask after its initial AgNPs load being fully characterized using atomic emission spectroscopy.

### 2.7. Determination of Efficiency against SARS-CoV-2

1 × 10^5^ PFU SARS-CoV-2 virus (40 µL of SARS-CoV-2 virus stock was added) was inoculated on the materials coated with silver nanoparticles and on their respective uncoated materials used as controls. SARS-CoV-2 virus was isolated from a COVID-19 patient at the Hospital Clínico Universitario Lozano Blesa (Zaragoza, Spain) [43]. Viral stocks were prepared and quantified using the epithelial cell line, Vero E6 (kindly provided by Julia Vergara from Centro de Investigación en Sanidad Animal IRTA-CReSA, Barcelona, Spain), as previously described in Santiago et al. [43]. Vero E6 cells were cultured in Dulbecco’s Modified Eagle Medium, obtained from Sigma-Aldrich (Darmstadt, Germany) and supplemented with 10% fetal bovine serum (FBS) (Sigma), 2 mM Glutamax (Gibco), 100 U/mL penicillin (Sigma), 100 µg/mL streptomycin (Sigma), 0.25 µg/mL amphotericin B (Sigma), 1% non-essential amino acids (Gibco), and 25 mM HEPES (4-(2- hydroxyethyl)-1-piperazineethanesulfonic acid; Biowest), referred as complete medium and used for cell expansion. Vero E6 cells were kept at 37 °C in a 5% CO_2_ humidified incubator and for the antiviral assays complete medium with 2% FBS was used. The materials inoculated with the virus were incubated for 10 min, 1 h, and 2 h at room temperature and subsequently the amount of infective virus was quantified as follows. Each sample was introduced into a 1.5 mL tube with 1 mL of 2% FBS medium and vortexed twice for 30 s. Six serial 1:10 dilutions were made, and 100 µL of each dilution was added in quadruplicate to Vero E6 cells seeded in a 96-well plate (10^4^ cells/well) and incubated for 72 h at 30 °C and 5% CO_2_. After the incubation time was completed, plates were fixed with 4% paraformaldehyde for 1 h at room temperature and stained with crystal violet, and the wells with viral cytopathic effect (CPE) were identified. The concentration of viable virus was determined by TCID50/mL. The 50% endpoint titers were calculated according to the Ramakrishnan simple formula (Equation (1)) [44], where $W_{CPE}$ refers to the total number of wells with CPE and $n_{W}$ to the number of wells per dilution.
(1)$$log_{10}\;of\;50\%\;endpoint\;dilution=\left(\frac{W_{CPE}\;}{n_{W}}+0.5\right)log_{dilution\;factor},$$

All procedures involving infectious virus handling were conducted in Biosafety Level 3 laboratories (BSL3). The SARS-CoV-2 virus stock was stored in screw cap micro tubes at −80 °C. The BSL3 facilities included a controlled air system with HEPA filters, working in under-pressure conditions to prevent air leakages and having an in-place inactivation of contaminated liquid waste. Among the personal safety measures, personal protective equipment (PPE) was worn, comprising disposable full body suits, arm sleeve covers, shoe covers, two pairs of gloves, and face masks. Additionally, samples that needed to be processed outside BSL3 facilities were successfully inactivated beforehand.

## 3. Results and Discussion

### 3.1. Silver-Nanoparticles Synthesis and Characterization

Using the original report of Goli et al. [41] as a reference, we studied the different synthesis variables that could have an effect on the resulting particle size distribution (Table 1), namely reagent concentrations, reducing agent concentration, and reagent addition rate. The reference NPs (from the original protocol) showed an average particle diameter of 12.3 ± 2.9 nm (Figure 2a). If the reagent addition was modified by adding them dropwise (Synthesis A-Table 1) to promote a controlled and fast mixing that enables an homogenous nucleation event, a particle size reduction (8.7 ± 2.0 nm) was observed (Figure 2b). In the syntheses where the addition of reagents was added in volumes of 1 mL (Synthesis D-Table 1) and 3 mL (Synthesis E-Table 1), the mean particle size increased by up to 11.2 ± 3.8 nm (Figure 2e) and to 13.4 ± 4.0 (Figure 2f), respectively. These results are in agreement with the literature [45] and confirm that the addition of reagents is a key variable in chemical reactions controlled by fast kinetics, such as the nanocrystallization of nanomaterials. On the other hand, if the concentration of reagents was halved to decrease the nanocrystallization rate and to control the growth event (Synthesis B), the mean particle diameter was reduced to only 6.2 ± 2.6 nm (Figure 2c). This phenomenon was also observed for Synthesis C (Table 1), where the concentration of reducing agent was halved and the mean particle size obtained was 8.0 ± 2.4 nm (Figure 2d). The larger mean size of nanoparticles resulted in synthesis C in comparison with B is rationalized by the high silver precursor concentration used in synthesis C. In this case, the nucleation event is similar to the one observed in synthesis B as it is confirmed by a similar size distribution, but the excess of Ag precursors promotes crystal growth and a resulting larger mean size was observed. If the concentration of Ag precursor and TSC was kept constant as in Synthesis B but the concentration of the reducing agent was increased, the particle size distribution was more heterogenous due to the fast and less controlled nucleation event (11.2 ± 3.8 nm, synthesis D, Figure 2e) and (9.8 ± 4.3 nm, Synthesis F, Figure 2g). When comparing Synthesis D and F, is it coherent that the mean size resulted in synthesis F was slightly smaller than that in D since the mixing process was also promoted by the dropwise addition of reagents and the fast stirring rate applied (1500 rpm).

As we mentioned before, the antimicrobial potential of silver nanoparticles is closely related to the morphology of the nanoparticles, being this action promoted with a high area per volume ratio, which corresponds with a rapid oxidation and a consequent fast ionic silver release rate [46]. An inverse dependence of the antimicrobial potential of silver nanoparticles has been described as a function of their size [47,48,49]. In all the selected samples used in this work, the morphology of the particles was typically spherical. Faced with this equality on the morphological conditions, it was decided to choose the synthesis that offered the smallest NP sizes (Synthesis B) for the subsequent studies aiming for a fast dissolution rate, ionic silver release, and a consequent rapid virus inactivation upon contact. In parallel, another desirable aspect of the silver nanoparticles obtained is the long-lasting stability of the resulting colloid. The UV-Vis spectra of the selected sample reported large stability over time when compared with the spectrum of the original synthesis. Specifically, syntheses A, B, and C demonstrated a narrower full width at half maximum of the silver-attributed absorbance peak (around 400 nm) associated with a reduced polydispersity on their composing particle sizes, which is in agreement with the TEM analysis and could benefit when obtaining a homogeneous coating.

### 3.2. AgNPs-PEI Based Coating Method and Microscopic Characterization

Lee et al. [39] described the combined antimicrobial use of AgNPs and PEI. In their work, the main action attributed to PEI was to stabilize the silver colloid by electrolyte-mediated electrostatic repulsion. They stated that the sustained release of silver ions was prolonged thanks to the protection of PEI on the silver particles surface, acting as a diffusion barrier and prolonging their action.

The high positive surface charge of PEI has been widely used to electrostatically interact with negatively charged surfaces [50,51]. One of the fundamental characteristics of this polymer is its theoretical ratio of primary, secondary, and tertiary amino groups when branched, established as 1:2:1, respectively [51,52]. As we mentioned before, the antibacterial activity of PEI has been widely described in the literature [39,53,54,55], both in its original form and in conjunction with nanostructured elements. However, to the best of our knowledge, there is still insufficient evidence of its effectiveness against SARS-CoV-2. Theoretically, its antibacterial activity is based on the strong supramolecular electrostatic interactions between this positively charged polymer and the negatively charged bacterial peptidoglycan, which is rich in carboxyl and amino groups. In our work, we took advantage of PEI cationic nature to facilitate electrostatic bonding to the surface of negatively charged polymeric fibers on the one hand and to negatively charged AgNPs on the other side. Specifically, it is presumed that the incorporation of PEI to the polymeric fibers used as substrate generates a highly positive film which remains sandwiched between the polymeric substrate and the AgNPs deposited on top during the coating process, taking advantage of its strong electrostatic and coordinate covalent bonding that amino groups form with metals as demonstrated in the sections below. This strong interaction between amino groups and metals was previously demonstrated by the presence of charge transfer, which led to a change in the protonation state of PEI [56].

During the optimization of the AgNPs–PEI binding method to the polymeric fibers, the concentration of AgNPs in the colloid and the proportion of PEI dissolved in water were varied. The objective of this optimization was to achieve a stable and uniform coating on polypropylene spunbond fibers used as substrates. In preliminary tests, PEI concentrations higher than 15 mg/mL were discarded due to the high solution viscosity, and therefore 5 and 10 mg/mL of PEI aqueous solutions were used in the subsequent studies. In the same way, the concentration of AgNPs in the colloid used in the fabric coating bath ranged from 1 to 5 mg/mL, as depicted in Figure 3. In all cases, even with the minor use of materials (5 mg/mL of PEI and 1 mg/mL of AgNPs), an efficient incorporation of silver on the polymeric fibers was observed. Silver loadings using 10 mg/mL of PEI and 5 mg/mL of AgNPs were used in the subsequent studies because of the even and homogenous coating achieved along the fabric surface and depth. EDX spectra obtained during SEM microscopy in random areas of the fibers showed the presence of Ag and C, attributable to the silver doping and to the polymer composition of the fibers, respectively.

With the established method, it was decided to coat other polymeric materials (Figure 4) based on polypropylene fibers, but with different configurations and morphologies, including polyester-viscose fibers in sewn fabrics and polypropylene fibers having glass fibers, used in the manufacturing of filtration media using the same conditions (e.g., 10 mg/mL of PEI and 5 mg/mL of AgNPs).

### 3.3. AgNPs-PEI Based Coating Stability Analysis under Different Boundary Conditions

Atomic emission spectroscopy (MP-AES) revealed a silver loading of ~0.5 mg/cm^2^ on spunbond fibers. The results of the stability analysis of the attachment under forced mechanical stress (Table 2) are shown in Figure 5. Samples sonicated in Milli-Q water showed an Ag loss of approximately 13 wt.% at 35 s increasing up to 30 wt.% at 45 s. However, in commercial silver textiles, nanosilver losses of up to 48–75% have been reported after water-based washing. Furthermore, those reports did not use ultrasonic sonication, but only conventional washes, so the presented method for binding silver to polymeric fibers represents a substantial improvement [57,58]. Slightly low binding results were obtained when using water at pH 7 as sonication media, where after 5 s of sonication the loss was 12 wt.% and, after 10 s the measured detachment increased up to 27 wt.%. AgNPs detachment using PBS (10%) were 8 wt.% and 24 wt.% after 5 and 10 s, respectively. Losses obtained after acidic and basic conditions were higher after sonication. At pH 4.5, around 70 wt.% of silver was lost, while at basic pH the loss was 42 wt.% (in the case of using pH 9) and 30 wt.% using pH 14. The acidification of the aqueous medium using HCl led to losses of more than 70 wt.% probably because the dissolution of silver is promoted under acidic conditions and the solubility of PEI increased [59,60,61,62]. Under alkaline conditions, the protonation of the amino groups is lost and therefore the interaction with the fabrics as well as with the AgNPs would vanish. However, these mechanical stress tests were much more restrictive than the product’s exposure under normal operating conditions. Sonication was used to challenge the strong chemical interaction and to prove that a strong attachment is present.

It was possible to quantify the loss of AgNPs from the coated fabrics after 30 manual washes in Milli-Q water thanks to a precision microbalance. Losses of 0.83, 0.44, 0.94, and 1.01 wt.% were measured in the spunbond of a coated surgical mask, a hygienic mask, a meltblown F9 fabric, and a HEPA H13 filter, respectively. Although the washing process is more aggressive than the exposure to air currents, it was decided to recreate this scene as it is foreseen in some standards related to the reuse of hygienic face masks (e.g., EN 0065). In essentially for none of the cases studied a loss greater than 1 wt.% was observed, which shows excellent stability results in Milli-Q water. This study was limited to water used at room temperature (not comparable to the conditions in a washing machine). However, after silver incorporation, fabric washing is required. If a wash is not carried out, the loss of particles can be overestimated since those particles with reduced adherence would be removed. In future works it will be necessary to evaluate the stability of the nanosilver attached on the filtering media against air flows. However, these studies were not considered in this work.

### 3.4. AgNPs-PEI Based Coating Antiviral Efficiency against SARS-CoV-2

The final goal of this work was the functionalization of common filter materials to provide them with virucidal ability against SARS-CoV-2. As shown in Figure 6a, the silver-coated materials achieve a large reduction in the viral load compared with the control materials (uncoated fabrics). The results indicated that after 2 h of infection, the materials reduced the viral load by more than 99.9%. Remarkably, within the first 10 min after infection, the viral load was reduced by more than 98% when deposited on spunbond fabrics (Figure 6b).

The role of PEI in the virucidal activity of the coating was also analyzed separately. The successful antiviral activity of this cationic polymer has been reported against several encapsulated viruses, such as influenza virus [63], hepatitis B virus [64], or human cytomegalovirus [65], and against non-encapsulated viruses such as human papillomavirus [65] or herpes virus [66,67]. Its primary mechanism of antiviral action is attributed to the blocking ability of the primary binding of the viral particles to proteoglycans receptors on target cells. However, the antiviral properties of PEI on SARS viruses have not been reported in detail [68].

A 10 mg/mL PEI spunbond coating was taken as a reference for antiviral testing in this work. In addition, 10- and 100-fold more diluted (1 mg/mL and 0.1 mg/mL) samples were prepared to analyze the effect of concentration (Figure 7a). Free PEI (in aqueous solution) was also analyzed to assess the exclusive activity of the polymer (Figure 7b). Thus, the effect of the substrate material (spunbond) was not analyzed during the second phase of the study. No decrease in the viral load was observed, so its virucidal activity was ruled out at the doses tested. All the antiviral ability of the coating was attributed to the AgNPs single activity. The study suggests that the capacity of PEI to inactivate SARS-CoV-2 is not efficient at the doses tested. In addition, PEI was toxic to Vero cells when incorporated in the coatings as well as in the aqueous solutions where it was used at a concentration of 10 and 1 mg/mL. No cytotoxicity was found at 0.1 mg/mL PEI concentrations and below (Table 3). In the coated samples using AgNPs and PEI, no cellular toxicity was observed at the doses tested.

## 4. Conclusions

The widespread use of masks to contain the COVID-19 disease is essential for the epidemiological management of the pandemic, even when individuals are vaccinated. However, the protection offered by surgical masks is compromised compared with the one offered by KN95 (FFP2) or KN99 (FFP3) masks. This paper reported a new antiviral coating that increases the antiviral efficiency of surgical masks and other types of masks (e.g., hygienic) and related materials used as air conditioning filters. The virucidal coating is based on the incorporation of AgNPs in the polymeric fibers. The addition of PEI during the bonding method is key to form a stable bond between the silver and the fibers. This cationic polymer has been reported to exhibit antiviral activity against enveloped and non-enveloped viruses. However, its effect on the SARS-CoV-2 virus was rarely discussed. In this work, we found that PEI did not affect the activity of the SARS-CoV-2 virus at the doses tested. In addition, we found cell toxicity from concentrations of 1 mg/mL onwards. The coatings based on AgNPs and PEI did not show cytotoxicity, although they did show a high viral inactivation efficiency. Two hours after infection, the viral load was reduced by more than 99.9% in all cases. However, part of the effect arises earlier, as at 10 min after infection, a 98% reduction of the viral load was already measured.

This work can be used to improve the epidemiological management of COVID-19. Using this method, the efficiency of respiratory protection equipment and air conditioning filtration systems can be improved. The binding of the silver particles to the fibers is stable. The method we present is easily scalable and with a low cost, which greatly facilitates its large-scale industrialization and production. However, it is still necessary to ensure the bond stability of the particles against air flows since these will be the real operating conditions.

## Figures and Tables

**Figure 1 materials-15-04742-f001:**
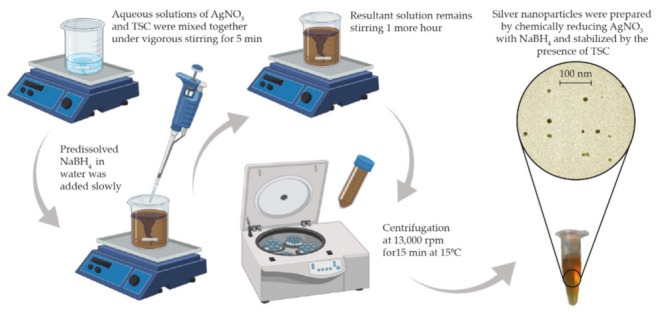
Schematic representation of the silver nanoparticle synthesis method.

**Figure 2 materials-15-04742-f002:**
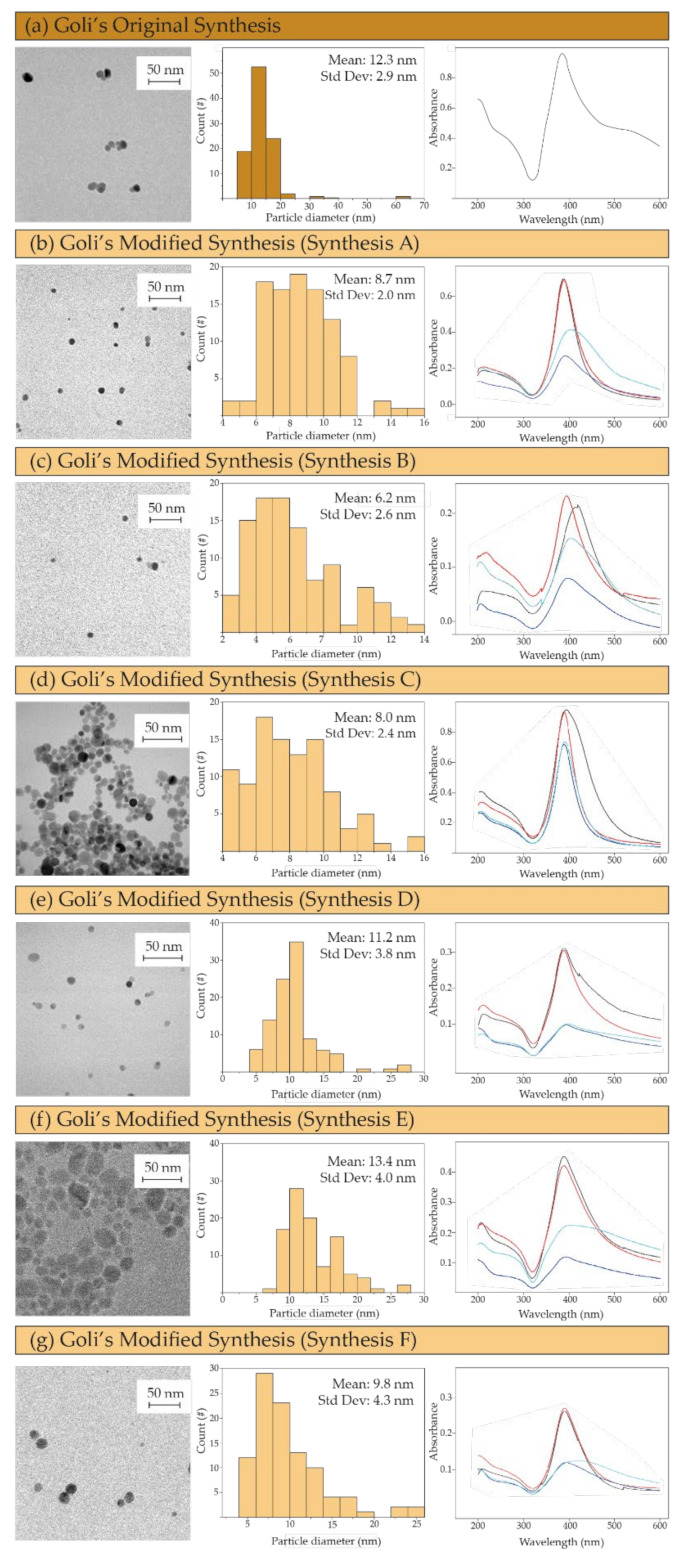
(**a**–**g**) TEM images of the silver nanoparticles obtained by following the different chemical syntheses together with their histograms showing the particle size distributions and their UV-Vis spectra on the first day of synthesis (black line) and at 3 (red line), 5 (blue line), and 7 days (green line) after synthesis. Particle size histograms were plotted after considering a population *n* = 100. Where # refers to the number of particles.

**Figure 3 materials-15-04742-f003:**
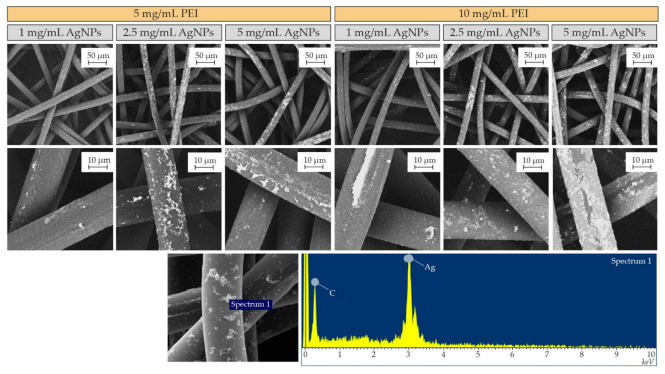
Backscattered electron SEM images of polypropylene spunbond fibers from a surgical mask coated with AgNPs–PEI, using two different PEI concentrations and three AgNPs colloid concentrations. The micrographs of the samples obtained with a higher amount of AgNPs and a higher PEI concentration show a significantly superior silver load. These images demonstrate that the concentration of AgNPs is as important as the amount of PEI to improve the silver incorporation yield. Bottom: EDX analysis on a selected coated area.

**Figure 4 materials-15-04742-f004:**
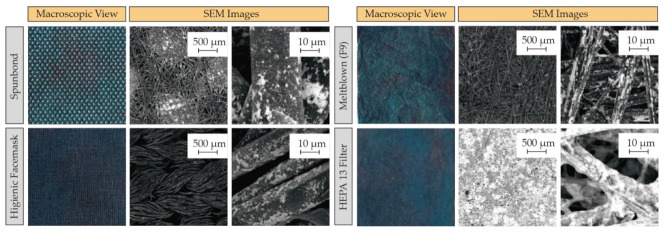
Macroscopic images and backscattered electrons SEM of different fabrics. A superior incorporation of AgNPs was observed in the fibers of smaller diameter (e.g., HEPA H13 filter). These images show that the silver load is homogeneous and high in all the polymeric samples studied (spunbond, hygienic facemask, meltblown, and HEPA filter).

**Figure 5 materials-15-04742-f005:**
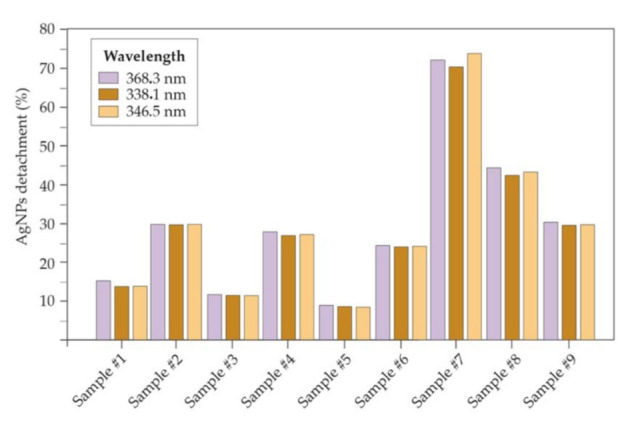
Percentage of AgNPs loss in each of the experiments performed with the AgNPs–PEI coating on polypropylene fibers (spunbond) observing three characteristic absorbance peaks of silver by atomic absorbance spectrometry (MP-AES).

**Figure 6 materials-15-04742-f006:**
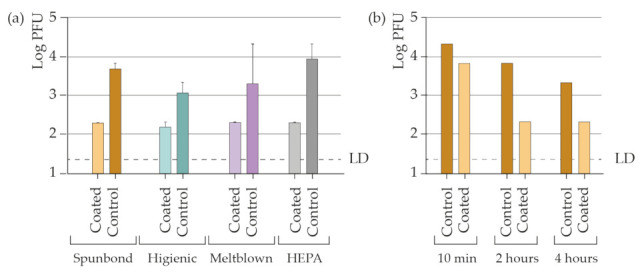
Determination of the infective viral load (**a**) in materials coated with silver nanoparticles and the control after 2 h and (**b**) in coated spunbond after 10 min, after 2 h after infection, and after 4 h after infection with the SARS-CoV-2 virus at room temperature.

**Figure 7 materials-15-04742-f007:**
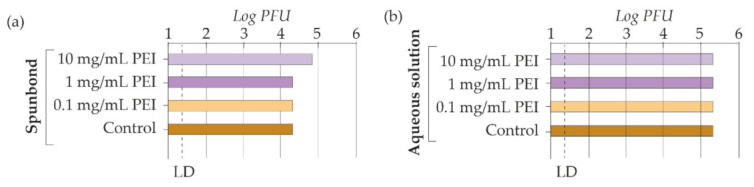
Determination of the infective viral load (**a**) in spunbond coated with PEI and (**b**) in PEI dissolved in milli-Q water 2 h after infection with the SARS-CoV-2 virus at room temperature. LD stands for limit of detection.

**Table 1 materials-15-04742-t001:** Optimized parameters from the original AgNPs synthesis.

Synthesis	Molar Concentration	Mixing Process	Stirring Speed
Original	2 mM AgNO_3_; 2 mM TSC; 2.4 mM NaBH_4_	Not defined	‘Vigorous’
A	2 mM AgNO_3_; 2 mM TSC; 2.4 mM NaBH_4_	Drop by drop	~1250 rpm
B	1 mM AgNO_3_; 1 mM TSC; 1.2 mM NaBH_4_	Drop by drop	~1250 rpm
C	2 mM AgNO_3_; 2 mM TSC; 1.2 mM NaBH_4_	Drop by drop	~1250 rpm
D	1 mM AgNO_3_; 1 mM TSC; 2.4 mM NaBH_4_	1 mL in 1 mL	~1250 rpm
E	2 mM AgNO_3_; 2 mM TSC; 2.4 mM NaBH_4_	3 mL in 3 mL	~1250 rpm
F	1 mM AgNO_3_; 1 mM TSC; 2.4 mM NaBH_4_	Drop by drop	~1500 rpm

**Table 2 materials-15-04742-t002:** Boundary conditions used for evaluating the release of AgNPs from coated polymeric substrates (polypropylene fibers from spunbond). All samples were initially sonicated in Milli-Q water to remove non-adhering particles. Some were sonicated again in other media, as indicated in the table. The second sonication medium varied from test to test.

Sample	Cleaning Sonication	First Sonication Conditions	Second Sonication Conditions
Reference	20 s in Milli-Q Water	-	-
1	20 s in Milli-Q Water	15 s in Milli-Q Water	-
2	20 s in Milli-Q Water	15 s in Milli-Q Water	10 s in Milli-Q Water
3	20 s in Milli-Q Water	15 s in Milli-Q Water	5 s in Tap Water (pH 7)
4	20 s in Milli-Q Water	15 s. in Milli-Q Water	10 s in Tap Water (pH 7)
5	20 s in Milli-Q Water	15 s in Milli-Q Water	5 s in PBS (10%)
6	20 s in Milli-Q Water	15 s in Milli-Q Water	10 s in PBS (10%)
7	20 s in Milli-Q Water	15 s in Milli-Q Water	10 s in Tap Water (pH 4.5)
8	20 s in Milli-Q Water	15 s in Milli-Q Water	10 s in Tap Water (pH 9)
9	20 s in Milli-Q Water	15 s in Milli-Q Water	10 s in Tap Water (pH 14)

**Table 3 materials-15-04742-t003:** Determination of the PEI toxicity in Vero cells at different dilutions.

	Percent of Toxicity (%)				
	Dilution 0	Dilution 1:10	Dilution 1:100	Dilution 1:1000	Dilution 1:10,000
Spunbond PEI coated (10 mg/mL)	100	100	0	0	0
Spunbond PEI coated (1 mg/mL)	100	0	0	0	0
Spunbond PEI coated (0.1 mg/mL)	0	0	0	0	0
PEI aqueous dissolution (10 mg/mL)	100	100	0	0	0
PEI aqueous dissolution (1 mg/mL)	100	0	0	0	0
PEI aqueous dissolution (0.1 mg/mL)	0	0	0	0	0

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
