# Peer review of "Silver Nanoparticles–Polyethyleneimine-Based Coatings with Antiviral Activity against SARS-CoV-2: A New Method to Functionalize Filtration Media"

_materials, 2022, doi:10.3390/ma15144742_

Round 1

Reviewer 1 Report

This manuscript can be accepted for publication after the authors provide sufficient responses to the following comments

1. Lines 122-123: for the synthesis of silver nanoparticles the authors used the method of Goli with slight modification, what is the modification conducted by the authors?

2. Lines 127-128: the authors mentioned that the colloidal have negative charged, did the authors measure it?

Author Response

This manuscript can be accepted for publication after the authors provide sufficient responses to the following comments

1. Lines 122-123: for the synthesis of silver nanoparticles the authors used the method of Goli with slight modification, what is the modification conducted by the authors?

Thank you very much for your recommendation. We appreciate your comment. Sorry if it was not clear in the initial version of our manuscript. We have optimized the original synthesis of Goli et al. by halving the concentration of AgNO3, NaBH4 and TSC and adding dropwise NaBH4. In this way, we have obtained particles with practically half the size of the one obtained with the original procedure. To improve the understanding, we have added now, in the revised version of the manuscript, a row in Table 1 (line 139) including the original synthesis parameters retrieved from Goli et al. We have also referenced that specific table (line 121).

2. Lines 127-128: the authors mentioned that the colloidal have negative charged, did the authors measure it?

Thanks for the question. We have not measured the zeta potential of our colloidal nanoparticles. However, it is well described in the original article published by Goli et al. and we have supported it with its corresponding reference.

Reviewer 2 Report

This paper can be publsihed with following major revision. 

1. Abstract should revised based on the obtained results

2.  Introduction is overexpressed if you consider the merit of the presented work against the current situation COVID19 and should be focused.

3. The necessity of two schemes is not justified. 

4.  TEM images supposed to be better in ethanol under ultrasonication where water allows more aggregation

5. Figure 3 should be revised to make it more visible and presentable 

6. Figure 4 needs more interpretation intsead of current form. 

7. Same for 5

8.  Last para of the introduction should be used to develop or propose a mechanism that could be a better way to lead this work. 

9.  Following  paper should be covered 

Antimicrobial silver nanoparticle-photodeposited fabrics for SARS-CoV-2 destruction, Colloid and Interface Science Communications, 2021,45,  100542

Nanomaterials 2021, 11(3), 581; https://doi.org/10.3390/nano11030581

Journal of hazardous materials, 2021, 408, 124919

Arabian journal of chemistry, 2019, 12, 908-931

Author Response

This paper can be published with following major revision.

1. Abstract should revised based on the obtained results.

Thanks for your suggestion. In the revised version of the manuscript we have disclosed the results obtained from the study. We have added the following text " Initially, the AgNPs synthesis procedure was optimized until reaching a particle size of 6.2 ± 2.6 nm, promoting a fast ionic silver release due to its reduced size, obtaining a stable colloid over time having reduced size polydispersity. The stability of the binding of the AgNPs to the fibers was corroborated using polypropylene, polyester-viscose, and polypropylene-glass spunbond mats, obtaining very low amounts of detached AgNPs in all cases. Under simulated operational conditions, a material loss less than 1% of nanostructured silver was measured. SEM micrographs demonstrated a high silver distribution homogeneity on the polymer fibers. The antiviral coatings were tested against SARS-CoV-2, obtaining inactivation yields greater than 99.9%." in the lines 25-32 of the Abstract. We hope that now most of the important results of the work are properly summarized in the abstract.

2. Introduction is overexpressed if you consider the merit of the presented work against the current situation COVID19 and should be focused.

We appreciate your comment. We have removed lines 39-58 from the initial version of the manuscript. We believe that this fact improves the focus of the introduction. We have also added additional explanations about the efficacy of commercial masks following reviewer's 3 suggestions.

3. The necessity of two schemes is not justified.

We agree with the reviewer. Accordingly, we have removed Scheme 2 (Figure 2) from the manuscript.

4. TEM images supposed to be better in ethanol under ultrasonication where water allows more aggregation.

We thank the reviewer for the suggestion. The reviewer is right but by using a solvent with a high vapor pressure such as ethanol, a non-constant evaporation rate  along the surface of the drop used for TEM sample preparation is promoted and consequently, the convective capillary flow induced within the drying drop would result in nanoparticle accumulation. This nanoparticle accumulation can also promote the aggregation if the drying process were too fast. It is recommended to balance the capillary flow and solvent evaporation rates. On the other hand, we did not transfer the Ag nanoparticles from water to ethanol to avoid any possible agglomeration process due to gravitational forces.

5. Figure 3 should be revised to make it more visible and presentable.

Thank you for your comment. We have redesigned Figure 3 (which is now Figure 2).

6. Figure 4 needs more interpretation intsead of current form.

Thank you for your suggestion. We have added in figure caption the following: "The micrographs of the samples obtained with a higher amount of AgNPs and a higher PEI con-centration show a significantly superior silver load. These images demonstrate that the concentration of AgNPs is as important as the amount of PEI to improve the silver incorporation yield." to improve the interpretation of figure 4 (which is now figure 3) (lines 339-341).

7. Same for 5.

Thank you for your comment. We have now added "These images show that the silver load is homogeneous and high in all the polymeric samples studied (spunbond, hygienic facemask, meltblown and HEPA filter)." to improve the interpretation of the figure 5 (which is now figure 4) (lines 346-347).

8. Last para of the introduction should be used to develop or propose a mechanism that could be a better way to lead this work.

We appreciate the suggestion but do not understand the reviewer's comment. We believe that the focus of our work is clear in the last paragraph of the introduction. We have included in the original version of the manuscript a proposed mechanism for the interaction between silver and the polymeric mats when saying “By suspending negatively charged polymeric fibers in this polymer, their charge is reversed, becoming positively charged. A strong supramolecular interaction by electrostatic interaction with negatively charged AgNPs is then guaranteed. In addition, the functional amine groups of PEI act as ligands and donate an unshared electron pair to Ag to create a coordinate covalent bond which further improves their strong attachment” (lines 102-107).

9. Following  paper should be covered:

Antimicrobial silver nanoparticle-photodeposited fabrics for SARS-CoV-2 destruction, Colloid and Interface Science Communications, 2021,45,  100542; Nanomaterials 2021, 11(3), 581; https://doi.org/10.3390/nano11030581 ; Journal of hazardous materials, 2021, 408, 124919. Arabian journal of chemistry, 2019, 12, 908-931

Thanks. We have added the most of the papers suggested by the reviewer as references [19], [33], and [37] (lines 65, 88, and 93) in the revised version of the manuscript.

Thank you again for your thoughtful suggestions.

Reviewer 3 Report

In this work, authors fabricated polypropylene spunbond fibers, polyester-viscose, and polypropylene-glass coated with silver nanoparticles and tested their use as antiviral protection layers towards SARS-CoV-2. Authors also performed optimization experiments on the synthesis of NPs and their adhesion to various polypropylene fibers with Polyethyleneimine as an adhesion. There are only a few to none grammatical mistakes, according to me, hence the article looks good can be considered to publish in the present form with few minor revisions.

1.     Line 57-58: This point can be further elaborated by providing further details and significant observations from the two article that are cited in this sentence for a better understanding.

2.     Line 63: The word “viricidal” to be replaced as “virucidal”.

3.     Line 88: Concise the observations form cited references on how the authors demonstrated the better antiviral behavior of Ag NPs than copper and zinc oxide.

4.     Line 98-105: Authors have rightly addressed the concern of inhalation of Ag NPs. Please cite the references which describe the adhesion of Ag NPs is guaranteed. Illustration with a picture describing the properties of polymer as well as AgNPs helping the adhesion could be more welcome.

5.     Line 138-140: “It was decided”. Could authors share an explanation or cite a reference for relation between size of the NPs to oxidation rate/ ionic silver release.

6.     Line 219:  Authors could elucidate the procedure to store SARS-CoV-2 virus stock. What were the precautions taken when working with these stocks?

7.     In figure 3(d) and 3(f), the NPs are seen as clusters compared to the NPs obtained in the other conditions. Could you please explain on this?

Author Response

In this work, authors fabricated polypropylene spunbond fibers, polyester-viscose, and polypropylene-glass coated with silver nanoparticles and tested their use as antiviral protection layers towards SARS-CoV-2. Authors also performed optimization experiments on the synthesis of NPs and their adhesion to various polypropylene fibers with Polyethyleneimine as an adhesion. There are only a few to none grammatical mistakes, according to me, hence the article looks good can be considered to publish in the present form with few minor revisions.

Thank you very much for the positive evaluation of our work and for your wise suggestions. We believe that the manuscript has significantly improved after introducing the proposed changes.

Line 57-58: This point can be further elaborated by providing further details and significant observations from the two article that are cited in this sentence for a better understanding.

In the revised version of the manuscript and according to your suggestions we have added " It has been reported that the use of masks has reduced up to 70% the chances of potential infections caused by SARS-CoV-2 [2][3][4]. However, the efficiency of surgical masks is compromised against submicron particle sizes [5], including SARS-CoV-2 loaded bio-aerosols [6]. Particle leaks between 60-80% have been reported when surgical masks as personal protective equipment, which could be explained mainly by the material filtering ability and the poor facial fitting [7][8][9][10]. Half masks are theoretically designed to provide a perfect facial fit, although in practice this is questionable [11][12][13]. A Centers for Disease Control and Prevention (CDC) report evaluating 21 KN95 masks on 25 volunteers revealed leaks between 6 and 88% of aerosolized particles [14]. In agreement with this results, Park et al. [15] reported leaks up to 21.1% in 3M KN95 masks and up to 73.3% in KF94 masks of Korean origin." to better understand these concepts in lines 43-53.

Line 63: The word “viricidal” to be replaced as “virucidal”.

Thank you for advising this misprint, which is now updated.

Line 88: Concise the observations form cited references on how the authors demonstrated the better antiviral behavior of Ag NPs than copper and zinc oxide.

Thank you for your suggestion. We added “Specifically, Merkl et al. [31] reported that silver nanoparticles reduced 98% of the SARS-CoV-2 viral load after 2 hours, while copper nanoparticles achieved only 76% reduction and zinc nanoparticles did not show antiviral efficacy” to clarify this concept in lines 83-86.

Line 98-105: Authors have rightly addressed the concern of inhalation of Ag NPs. Please cite the references which describe the adhesion of Ag NPs is guaranteed. Illustration with a picture describing the properties of polymer as well as AgNPs helping the adhesion could be more welcome.

We appreciate your comment. To absolutely guarantee the adhesion of the AgNPs, complementary studies (based on international regulations) are required, which we cannot address in this work. Nevertheless, we indicate that our preliminary studies of particle detachment in liquid media are promising as we report in the manuscript.

We have used the figure with the description of the nature of the chemical bonding between the silver nanoparticles and the polymeric mats to illustrate the Graphical Abstract.

Line 138-140: “It was decided”. Could authors share an explanation or cite a reference for relation between size of the NPs to oxidation rate/ ionic silver release.

Thank you. We have now added [22] and [45] references in line 138 as recommended.

Line 219:  Authors could elucidate the procedure to store SARS-CoV-2 virus stock. What were the precautions taken when working with these stocks?

Thank you for your concern. We have added in the revised version of the manuscript the following: "All procedures involving infectious virus handling were conducted in Biosafety Level 3 laboratories (BSL3). The SARS-CoV-2 virus stock was stored in screw cap micro tubes at -80 ºC. The BSL3 facilities included a controlled air system with HEPA filters, working in under-pressure conditions to prevent air leakages and having an in-place inactivation of contaminated liquid waste. Among the personal safety measures personal protective equipment (PPE) were worn, comprising disposable full body suits, arm sleeve covers, shoe covers, two pairs of gloves and face masks. Additionally, samples that needed to be processed outside BSL3 facilities were successfully inactivated beforehand." in  the Methods (lines 238-245).

In figure 3(d) and 3(f), the NPs are seen as clusters compared to the NPs obtained in the other conditions. Could you please explain on this?

The reviewer rise an interesting observation. Thank you. The convective capillary flow induced within the drying drop from the center of the droplet towards its edges results in a net transport of dispersed nanomaterial to the edge, which is known as the coffee-ring effect [Deegan R. D. et al.: Nature 389, 827-829 (1997)]. The deposition of the nanoparticles near the edge of the droplet can cause substantial aggregation. This effect is a major cause of the non-uniform deposition of nanoparticles during TEM sample preparation. However, in our case we select these areas while TEM imaging because the number of nanoparticles was high enough to perform statistic studies (particle size distribution histograms) but there were other areas with few nanoparticles where no aggregation was observed (coffee-ring effect).